# Phase Change Material (PCM) Composite Supported by 3D Cross-Linked Porous Graphene Aerogel

**DOI:** 10.3390/ma15134541

**Published:** 2022-06-28

**Authors:** Chengbin Yu, Young Seok Song

**Affiliations:** 1Research Institute of Advanced Materials (*RIAM*), Department of Materials Science and Engineering, Seoul National University, Seoul 08826, Korea; ycb0107@snu.ac.kr; 2Department of Fiber Convergence Materials Engineering, Dankook University, Yongin-si 16890, Korea

**Keywords:** PCM composites, graphene aerogel, volume shrinkage, pyro system, energy harvesting

## Abstract

Integration of form-stable phase change material (PCM) composites with a pyro system can provide sufficient electrical energy during the light-on/off process. In this work, modified 3D porous graphene aerogel is utilized as a reliable supporting material to effectively reduce volume shrinkage during the infiltration process. Poly(vinylidene difluoride) (PVDF) is used for a transparent pyro film in the pyro system. The temperature fluctuation gives rise to a noise effect that restricts the generation of energy harvesting. The cross-linked graphene aerogel consisting of PCM composites can stabilize the temperature fluctuation in both melting and cooling processes. This shows that PCM composites can be applied to the pyro system under the change of the external environment. To evaluate the experimental results, a numerical simulation was conducted by using the finite element method (FEM).

## 1. Introduction

Solar-to-electrical energy harvesting is a general utilization for practical applications. External solar energy is absorbed to store as thermal energy, which can be transferred into several forms of renewable energy to reduce environmental pollution [1,2]. The thermoelectric power generator (TEG) is composed of working materials to induce electron movement [3,4]. The TEG with PN junctions is an effective device to generate induced electrical energy under the temperature variation of the functional device by the Seebeck effect [5,6]. Long-term temperature variation leads to usable pyro-to-electrical conversion to provide large external energy. To achieve thermal energy accumulation, phase change materials (PCMs) are normally used due to their high latent heat and thermal properties [7,8]. Multiple PCMs have been introduced to make a recyclable thermoelectric energy harvesting system that can generate electrical energy during the phase transition process [9,10]. A great advantage of multiple PCMs is that they can change the temperature automatically and quickly and effectively produce electrical energy [11,12].

However, a PN TEG-based power generator requires high costs that may restrict further applications, e.g., film-like electrodes. To increase the normal utilization of the energy harvesting system, pyroelectric technology employs a smart power generator during temperature change conditions [13,14]. The component materials of widely utilized pyro-electrode are poly(vinylidene difluoride) (PVDF), BaTiO_3_, metal materials, and/or indium tin oxide (ITO) glass [15,16]. It is easy for the pyro-electrodes to generate electrical energy without any special conditions [17,18]. Much research regarding pyroelectric energy harvesting has been reported from using the absorption of solar irradiation [19,20]. To provide stable and continuous electrical voltage and current, the main materials should be placed on the pyro-electrode [21,22]. Therefore, PCMs are the most appropriate candidates to achieve the main purpose of energy harvesting [23,24]. The high thermal energy storage (TES) is applied to the pyro system by using PCM composites, which can make a stable temperature variation and generate stable pyroelectric energy in the closed circuit [25,26].

The form-stable PCM composites that are able to be fabricated without any leakage problem are regarded as a thermal reservoir to absorb or release sufficient heat energy because of high latent heat (ΔH) [27,28]. Hence, there are many reports about reduced graphene oxide (rGO)-supported nanosheets and electro-mode, which can sustain initial circumstance [29,30]. Graphene aerogel is selected to hold pure PCM for sustaining the initial solid state without any leakage for pyro-to-electrical energy harvesting [31,32]. Considering the volume shrinkage during the infiltration process, the high mechanical property induced by graphene aerogel can increase the total TES capacity of PCM composites [33,34]. For a transparent pyro system, external solar light can transmit easily into the glass and PVDF films and increase the temperature on the other side of pyro-electrode [35,36]. The opposite window glass of the pyro-electrode starts to absorb the transmitted sunlight and store it as a heat resource during the light-on process. Polarization from the pyro film can generate the pyroelectric energy harvesting, and the control of the temperature fluctuation can provide a stable and continuous output voltage and current [37,38]. The pyro-electrode can generate a usable electrical energy until the completion of PCM melting process. When the light source is removed, the temperature close to the solar light decreases rapidly. However, the temperature of the PCM composite changes slowly due to the existence of the PCM during the light-off process.

In this work, we constructed a sustainable pyro-to-electrical power device that is composed of the pyro system and PCM composites. The temperature variation in the window glass is dependent on the light-on process, due to the different light intensity. To observe the pyroelectric effect, we used the multiple PCM composite system with a copper/PVDF pyro-electrode and ITO/PVDF transparent power generator. This system can easily control stored thermal energy by using PCMs and convert the pyroelectric energy during the absorption of solar light. With the smart control of transparent pyro-electrode, the PCM composite releases the stored heat to generate electrical voltage and current during cooling process. PEG or 1-TD composites are utilized as a sunlight blind and exhibit great thermal reliability under the duration of pyro-to-electric energy harvesting. This study briefly indicates a potential application of smart solar energy harvesting by using PCM composite.

## 2. Experimental Sections

### 2.1. Materials

For fabrication, we used multiple phase change materials (PCMs), polyethylene glycol (PEG Mn = 6000; Avention^®^ Corporation, Siheung-si, Korea), and 1-tetradecanol (1-TD; Sigma–Aldrich, St. Louis, MO, USA). The powder of graphite, graphene nano-platelet (GNP, C grade), potassium permanganate (KMnO_4_), sulfuric acid (H_2_SO_4_), cysteamine monomer, nitric acid (HNO_3_), citric acid, and perchloric acid (HClO_4_) were applied to fabricate a cysteamine cross-linked graphene aerogel (GCA).

### 2.2. Fabrication of Final Composite

For the fabrication of PCM composites, the first step is to increase the amounts of oxidized groups on the surface of graphene skeleton. The prepared graphene oxide (GO) powder, by following the instructions of the modified Hummers’ method [39,40], is mixed with the 90 mL nitric acid, at 275 rpm, stirring for 5.5~6 h over 120 °C [41]. The KMnO_4_ solid powder is dispersed sufficiently in DI water and ready to mix with GO&HNO_3_ for over 3 h, under the oil bath. Then perchloric acid (HClO_4_) is utilized to increase the GO oxidation, and the citric acid aqueous solution can neutralize the mixture to remove the excess oxidizer. The last oxidized GO is entitled GO&KMnO_4_, which can be fabricated by the freeze-drying method to obtain the graphene aerogel.

The fabrication of KMnO_4_ oxidized GO aerogel by adding graphene nano-platelet (GNP) can improve the thermal morphology due to the 3D structure. The oxidized GO and GNP fillers are poured into a 4 cm × 4 cm × 0.5 cm mold, and the solvent is evaporated by the freeze-drying method for over 48 h. When the cysteamine aqueous solution is completed, all of specimens are put into the vacuum chamber for 72 h (150 °C). In the chamber, the cysteamine gas is evaporated to reach the oxidized GO aerogel internal skeletons, and the cross-linked cysteamine is reacted. In this way, the cysteamine reacted graphene aerogel (GCA) is prepared and used. Both PEG and 1-TD composites are manufactured by inserting pure PCMs into the GCAs. These multiple PCMs were changed into the liquid phase to infiltrate into supporting materials of GCAs under vacuuming process for at least 6 h. The cross-linked GCA-supported PCM composites exhibit high mechanical property that can bear external force without any leakage problem.

### 2.3. Construction of Pyro System Device

Decreasing the temperature fluctuation is necessary to generate pyro-to-electrical energy harvesting, and it is required to prevent temperature loss in the pyro-electrode. Therefore, the PCM composites are connected to the pyro-electrode, and the change of temperature is controlled during the phase transition process. Figure 1 shows a schematic of the pyro-electrode with PCM composites for solar-to-electrical energy harvesting. According to the principle of pyroelectric energy harvesting, the temperature difference between two sides of the electrode causes temperature fluctuation, which could generate an electrical current in the circuit. Multiple PCM composites are employed to connect with a pyro-electrode. The output electrical current was obtained under the change of external temperature due to the different phase transition fields [42].

The transparent pyroelectric energy harvesting device can transmit solar light into the pyro system. The cross-linked GCA composed PCM composites has connected to the one side of the pyro device to absorb the solar light during the PCM melting process [43]. The solar light forward to the pyro system is fixed to (10 and 15) mW/cm^2^. The window glass toward the solar light becomes a hot side, and the pyro system with the PCM composite undergoes pyro-to-electrical energy movement due to the temperature variation under two sides of pyro device. When the light source is removed, the PCM-connected pyro system starts to release plenty of stored heat, and during the PCM cooling process, the pyro-to-electrical energy harvesting is still generated. The window glass connected to PCM composite exhibits a hot side, and during the PCM phase transition, the pyro-electrode can provide additional electrical energy conversion. These two intensities of solar light are required for an appropriate phase transition field, and multiple PCM composites are selected at different conditions.

### 2.4. Characterizations

The synthesized GCA peaks were measured by Fourier-transform infrared spectroscopy (FTIR, Varian 660, UT, USA), and Raman spectroscopy (LabRAM HR Evolution, HORIBA, Kyoto, Japan) was provided to confirm the reaction of GCA. The graphene aerogels stress-strain curves were obtained by Universal Testing Machine (UTM, Instron, Norwood, MA, USA). The field-emission scanning electron microscopy (FE-SEM, Merlin compact, ZEISS, Germany), with 5 kV accelerating voltage, was used to observe the surface morphologies of GCA and PCM composites. For the phase transition temperature and latent heat (∆H) between the pure PCM and PCM composites, differential scanning calorimetry (DSC4000, PerkinElmer, Waltham, MA, USA) was utilized under the temperature range from initial 15 °C to final 90 °C, at a rate of 10 °C/min. The typical peaks between the GCA and PCM composite were observed by X-ray diffraction (XRD, New D8, Bruker, Billerica, MA, USA), at a rate of 3°/min, from (2θ) 10 to 60°. A thermal analyzer (C−Therm TCI, C−Therm Technologies Ltd., Fredericton, NB, Canada) with a modified transient plane source method was selected to measure the thermal conductivities of the PCM composites. Solar light (Ultra-vitalux 300 W, OSRAM, Munich, Germany) was utilized as a light source, and a functional multi-meter (UT61, UNI-T, Dongguan, China) could measure the output electrical energy during the light-on/off process. To construct a pyro-electrode, poly(vinylidene difluoride) film (PVDF, MEAS, THOMAS, University Place, WA, USA) was connected to copper film and indium tin oxide (ITO) to harvest pyroelectric energy.

## 3. Numerical Analysis

COMSOL Multiphysics, one of the popular software programs with finite element method (FEM) process, was employed for calculating the deviation of temperature variation during the melting and cooling processes [44]. The phase change material (PCM)-composites-constructed pyro-electrode could generate solar-to-electrical energy harvesting due to the temperature variation from the two sides of the pyro device. This controllable pyro-electrode was modeled by deciding on the PCM composites, window glass, ITO glass, copper film, pyroelectric film (PDVF), and external intensity. The mesh elements utilized in this simulation were over 31,000, and the output electrical energy was generated by change the temperature profiles. The basic governing equation of heat transfer is described below:(1)$${\rho C}_{p}\frac{\partial T}{\partial t}+{\rho C}_{p}u\cdot \nabla T+\nabla \cdot q=Q$$
where ρ is the mass density, $C_{p}$ is the heat capacity, and q shows the heat transfer rate. Because q=−k∇T refers to the thermal conductivity, the density of PCM (ρ) is proportional to solid and liquid phase ratios, as shown below:(2)$$\rho ={\theta \rho }_{\mathrm{phase}1}+\left(1-\theta \right)\rho_{\mathrm{phase}2}$$
where θ represents a PCM fill factor, and the equations that refer to heat capacities are connected to the fill factor, θ, as shown below:(3)$$C_{p}=\frac{1}{\rho }\left({\theta \rho }_{\mathrm{phase}1}C_{p.\mathrm{phase}1}+\left(1-\theta \right)\rho_{\mathrm{phase}2}C_{p.\mathrm{phase}2}\right)+L\frac{\partial \alpha_{m}}{\partial T}$$

The PCM composites’ thermal conductivities are the total value of the contributions between solid and liquid phases, as shown below:(4)$$k={\theta k}_{\mathrm{phase}1}+\left(1-\theta \right)k_{\mathrm{phase}2}$$

Equation (5) shows the solid and liquid phases of mass coefficient during temperature variation, and the BC condition is given as $-n\cdot q=q_{0}$.
(5)$$\alpha_{m}=\frac{1}{2}\frac{\left(1-\theta \right)\rho_{\mathrm{phase}2}-{\theta \rho }_{\mathrm{phase}1}}{{\theta \rho }_{\mathrm{phase}1}+\left(1-\theta \right)\rho_{\mathrm{phase}2}}$$
where n is a normal vector, and $q_{0}$ is presented as the heat flux under the solar light with a melting process. Natural cooling starts when the light source is removed. The heat flux, $q_{0}$, is related to the heat transfer coefficient $\left(h_{\mathrm{air}}\right)$ and the temperature difference, as shown in Equation (6). The external temperature, $T_{\mathrm{ext}}$, is correlated to the total thermal transformation among the PCMs’ melting and cooling processes.
(6)$$q_{0}=h_{\mathrm{air}}\cdot \left(T_{\mathrm{ext}}-T\right)$$

The equations for the output pyro system are composed of output voltage and induced current, as show in Equations (7) and (8) below:(7)$$V=\frac{p^{*}}{\varepsilon }h∆T$$
(8)$$I=p^{*}A\cdot ∆T/\mathrm{dt}$$
when $p^{*}$ is the pyroelectric coefficient, ε is the polarization permittivity, and h represents a thickness of the pyro film; all of these key factors are easy to calculate the pyro-to-electrical energy movement. The induced current from the pyro system refers to the surface area of the pyro system and the change of temperature variation, ∆T/dt. The value of ∆T/dt in this study is described as $∆T/\mathrm{dt}=\left(T_{n+1}-T_{n}\right)/\left(t_{n+1}-t_{n}\right)$. The change of temperature variation (∆T/dt) under the pyro system is shown below:(9)$$∆T/\mathrm{dt}\{\begin{array}{c} >0\;\;\;\;\;\;\;\mathrm{Increase}\;\mathrm{of}\;\mathrm{heating}\;\mathrm{or}\;\mathrm{decrease}\;\mathrm{of}\;\mathrm{cooling}\;\;\;\;\;\;\;\;\;\;\\ =0\;\;\;\;\;\;\mathrm{No}\;\mathrm{pyroelectric}\;\mathrm{current}\;\;\;\;\;\;\;\;\;\;\;\;\;\;\;\;\;\;\;\;\;\;\;\;\;\;\;\\ <0\;\;\;\;\;\;\;\mathrm{Increase}\;\mathrm{of}\;\mathrm{cooling}\;\mathrm{or}\;\mathrm{decrease}\;\mathrm{of}\;\mathrm{heating}\;\;\;\;\;\;\;\;\;\; \end{array}$$

From Equation (9), we can see that the ∆T/dt is a key factor to measure the output-induced current under the melting and cooling processes. Meanwhile, the pyro system’s generated voltage can be shown as V=I·R. The R is the resistance of the pyro system, and the pyro-to-electrical energy harvesting efficiency, η, is a function of heat energy, Q, and η=W/Q, which relate to the transmitted thermal energy, W.

## 4. Results and Discussion

### 4.1. The Characteristics between PEG and 1-TD Composites

Figure 2a shows the GO&KMnO_4_ aerogels and the cross-linked graphene aerogels, as well as the cross-linked graphene/cysteamine aerogels (GCAs) turned a black color. Figure 2b shows the characteristic peaks of GO functional groups. The stretching vibration peak of O−H is in the vicinity of 3400 cm^−1^, C = O stretching vibration peaks are at around 1700 cm^−1^, and the C−O stretching peak is at around 1100 cm^−1^, which disappears after the reduction of GO. The two new peaks that appear at 750 and 1250 cm^−1^ demonstrate the generation of C−S and C−N curves. Figure 2c shows the Raman peaks that were mentioned to further demonstrate the oxidation degree of GO. The GO&HNO_3_ exhibited a higher intensity than that of the pristine GO, which showed the additional generation of functional groups on the graphene layer. The GO&KMnO_4_ had the highest intensity in both the D and G peaks and was decreased rapidly by the cysteamine treatment. The intensity ratio of I_D_/I_G_ of GO was 0.90 and was slightly increased to 0.94 at the GO&HNO_3_. Furthermore, the GO&KMnO_4_ had a 1.01 intensity ratio, and GCA turned into 1.08 at the end of the reduction. The Raman peaks indicated the importance of nitric acid, which provided more functional groups to oxidize the carboxyl structures [45]. Figure 3a shows the shape recovery test, and Table 1 gives the results. Both GCA 1:1 and GCA 1:2 recovered to their initial shapes by nearly 100%, and they briefly demonstrated that GCA can effectively prevent volume shrinkage. To demonstrate the mechanical properties, the stress–strain curves were obtained to further demonstrate the increase of mechanical properties, as given in Figure 3b. The samples were compressed at the ε = 50% maximum strain and showed that, during the compression process, the initial graphene aerogels (GAs) exhibited a weak mechanical property. The GCAs showed excellent mechanical properties, with over 3.0 kPa stress, which indicated a significant increase in the cross-linked internal structures. The KMnO_4_ created more oxidized functional groups to combine with cysteamine and showed a more rigid structure than that of the without-oxidation graphene aerogels. Figure 4 shows the shape stability of PEG and the 1-TD composites that were observed via a leakage test. The PCMs were placed on the hot plate, and the temperature was increased from 25 to the final 80 °C. The camera images revealed that all of the graphene-aerogel-supported PCMs sustained the solid state under the temperature variation. Hence, during the phase transitions, the PCM composites formed stable shapes, without any leakage. When the mechanical test began with the external force (2 N) at 80 °C, the original graphene-aerogel-supported PCM composites showed leakage due to the weakness of their mechanical properties. However, the GCA-supported PEG and 1-TD composites exhibited high tolerance, without any volume leakage; thus, the GCA-supported PCM composites demonstrated well that they can maintain the intrinsic solid state at the change of external conditions. 

Figure 5a,d shows the DSC measurement of the PEG/GCA and 1-TD/GCA composites, respectively. Table 2 lists the characteristics of temperature and thermal enthalpy. The melting temperature (T_m_) of PEG/GCA was obtained at 64.31 °C, and the latent heat (ΔH) was 179.65 J/g [46]. The 1-TD composite’s melting point appeared at 42.76 °C, and the latent heat (ΔH) was measured as 214.01 J/g. For the cooling process, the PEG composite’s cooling point (T_c_) was 39.19 °C, and the enthalpy was 159.68 J/g. On the other hand, the 1-TD composite had 30.42 °C, and the cooling enthalpy was 211.94 J/g. According to the DSC results, both the PEG and 1-TD composites were utilized to construct the thermoelectric energy harvesting system, due to their high thermal energy storage. Figure 5b,e shows the DSC thermal cycling peaks of the PEG and 1-TD composites. After the 100 cycling tests, both the PEG and 1-TD composites exhibited similar peaks and maintained their intrinsic high thermal enthalpies. The results of the FTIR peaks indicated that no new peaks appeared at the PCM composites, as shown in Figure 5c,f. From the DSC cycling and FTIR tests, the PCM composites exhibited excellent thermal reliabilities and chemical stabilities to be utilized for pyroelectric energy harvesting. Figure 6a,b shows the SEM images of the GCAs. There were net-like wrinkles around the internal skeletons and indicated the combination of cross-linked molecular chains that were generated completely. The GCAs had high porosities that could infiltrate pure PCMs to fabricate the form-stable PCM composites. Figure 6c shows that the PEG was held in the GCA when the 1-TD composite was given in Figure 6d. The PCM composites without GNP merely showed 0.34 and 0.26 W/mK, respectively, while the GNP-embedded PCMs were significantly increased to 0.58 and 0.41 W/mK, as shown in Figure 6e. This indicated that the GNP fillers could increase the thermal conductivities of PCM composites. Figure 6f presents the weight fractions of PEG and 1-TD. The two different PCMs showed high TES capacity that was close to 98% during the phase transition process. Figure 6g shows the XRD peaks of GCA, PEG/GCA, and 1-TD/GCA. From the sharp peaks of each PCM composite, we can see that there were no chemical reactions between GCA and the pure PCMs.

### 4.2. PEG and 1-TD Composites Pyro System

The formula showed that, during the light-on/off process, the output current was dependent on the output voltage from the pyro-electrode. However, the pyro-electrode without PCM composite had irregular output voltage and current, as shown in Figure 7a,b, because the temperature fluctuation generated electrical noise and, thus, could restrict further applications. Figure 7c,d shows the output voltage and current by connecting the PCM composites. The electrical energy was generated during the light-on/off process by the existence of the temperature difference between the two sides of the pyro-electrode. The maximum output voltage and current were approximately 140 mV and 1.4 μA, respectively. The pyroelectric system with multiple PCM composites can effectively control the temperature fluctuation to generate a stable and continuous output voltage and current during the light-on/off process.

Figure 8a shows the temperature profiles under 10 mW/cm^2^ solar light intensity. While the solar light intensity was increased to 15 mW/cm^2^, both the PEG and the 1-TD composites finished the phase transition process effectively, as presented in Figure 8b. After connecting the PCM composite, the temperature profiles were generated continuously during the light-on/off process. However, it was hard to proceed with the phase transition of the PEG composite under relatively low light intensity, while the 1-TD composite could absorb sufficient solar light during the light-on process. The maximum temperature difference was over 30 °C. Therefore, the 1-TD composite was connected to the transparent pyro-electrode to generate the electrical energy. For 15 mW/cm^2^, Figure 8d shows the temperature difference between the PEG and 1-TD composite. To compare the two PCM composites temperature peak areas, Figure 8e presents the calculated peak ratios, and the PEG-composite-connected pyro-electrode shows excellent temperature variation under 15 mW/cm^2^. The 1-TD composite’s change of temperature difference was measured successfully, as shown in Figure 9a. It was fully demonstrated that the change of temperature difference can produce a pyro effect and provide the output electrical current. Figure 9b,c shows the output voltage and current during the light-on/off process under 10 mW/cm^2^. In addition, Figure 9d shows the change of temperature difference by connecting PEG composite under 15 mW/cm^2^. The peak profiles in Figure 9e,f show that, during the light-on/off process, both a stable output voltage and current were successfully obtained.

### 4.3. Simulation Date of Pyro-To-Electrical Result

The output curves in both the experimental and numerical results are illustrated to estimate the pyro energy harvesting by controlling the temperature. Figure 10a shows the temperature gradients of the PEG and 1-TD composites during the light-on process. The 1-TD peak is increased slowly due to the low melting point at the first time. However, the PEG starts the solid–liquid phase transition process to reduce the temperature difference and even reverse from 1500 s. After removing the light source, the temperature decreased to nearly room temperature, as shown in Figure 10b. The temperatures of both PEG and 1-TD decreased rapidly. PEG was under crystallization process, which is faster than that of 1-TD. In regard to the temperature difference between two kinds of PCMs, Figure 10c shows that the maximum temperature difference during the light-on/off process was approximately (16 and 12) °C, respectively. To generate an induced current, the temperature variation is necessary, as demonstrated in Figure 10d. It was found that the numerical results predicted the output profile of the pyroelectric effect successfully under the phase transition process between PEG and the 1-TD composites. Figure 10e presents the measured and simulated output voltage curves during the light-on/off process. The simulation results of output voltage were quite similar to the experimental profiles, and this can further demonstrate the accuracy of pyroelectric energy harvesting under the PCM composites melting process. Figure 10f compares the peaks of output current during the light-on/off process, and these results exhibit great agreement under the PCM composites’ cooling process. Figure 10g shows the output electrical harvesting efficiency during the light-on/off process, and the pyroelectric energy harvesting efficiencies were closed to 67.81% and 41.36%, respectively. The pyroelectric device with multiple PCM composites played a potential development in the solar-to-electrical energy harvesting applications.

For the transparent pyro-electrode, the calculation results, including the temperature variation, output electrical results, and efficiencies under solar irradiation, were demonstrated. Figure 11a shows the glass and 1-TD composite temperature profiles under a light intensity of 10 mW/cm^2^. The glass temperature was increased rapidly under the light-on process, and the 1-TD composite absorbed the solar light effectively to store the thermal energy. When the light intensity was changed to 15 mW/cm^2^, the glass temperature increased to 78 °C, and the PEG composite fully completed the phase transition process, as shown in Figure 11b. The temperature profiles indicated that the numerical simulations matched the experimental results during the light-on/off process. Figure 11c shows the profiles of temperature difference under the (10 and 15) mW/cm^2^ solar light intensities. Both the PEG and 1-TD composites had large peak areas to generate pyroelectric energy harvesting by the change of temperature difference, as shown in Figure 11d. The simulation peaks were quite similar to the experimental results, suggesting the possibility of a pyro effect during the PCM composites phase transition process. Figure 11e,f illustrates the value of output voltage and current during the light-on/off process. The maximum voltage was over 220 mV due to the change of temperature difference between the two sides of the pyro-electrode. The stable and continuous current from all the results showed that the PCM-composite-connected transparent pyro-electrode could provide usable electrical energy by absorbing the solar light. The pyroelectric energy harvesting efficiency under 10 mW/cm^2^ showed 58.67 and 45.12%, respectively, as seen from Figure 11g. After the intensity was increased to 15 mW/cm^2^, the harvesting efficiency during the light-on/off process was 57.93 and 46.22%, as shown in Figure 11h. The transparent pyro-electrode indicated new prospects for solar-to-electrical energy harvesting applications.

## 5. Conclusions

In this study, we fabricated a new platform for a pyroelectric power generator to produce solar-to-electrical energy harvesting. The multiple-PCM-composites-constructed pyro-electrode and ITO transparent glass of the pyro-electrode were utilized effectively for the smart control of pyroelectric energy harvesting. The multiple-PCM-composites-connected pyro-electrode showed stable and continuous output voltage and current during the light-on/off process. Based on the different solar light intensities, the 1-TD composite provided a more appropriate PCM blind than that of the PEG composite under the 10 mW/cm^2^ solar light intensity. It was hard to achieve the phase transition field in the PEG composite, due to the low solar light intensity. When the solar light intensity was increased to 15 mW/cm^2^, both the PEG and 1-TD composite exhibited a large increase of temperature ranges during the light-on/off process. The PEG composite was selected as a PCM blind, due to the high temperature peak ratio. The pyro-electrode of window glass can provide an opportunity for the smart control of solar-to-electrical energy harvesting applications.

## Figures and Tables

**Figure 1 materials-15-04541-f001:**
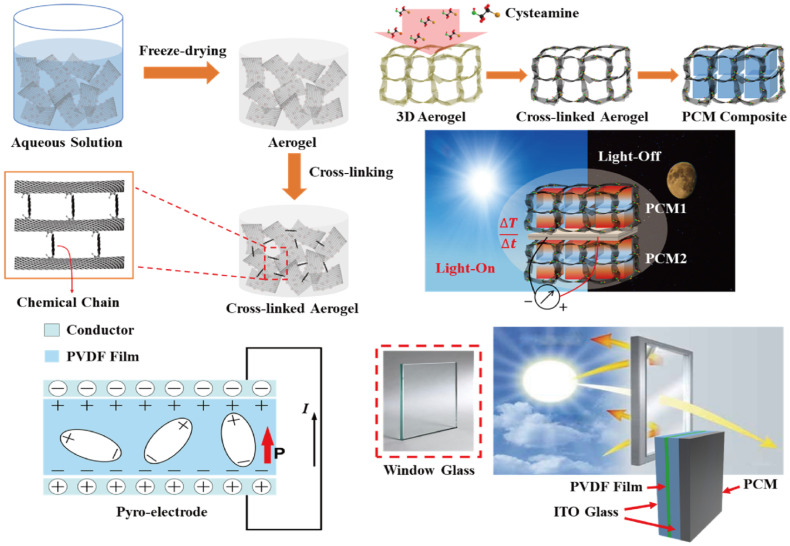
Schematic of the fabrication of cross-linked graphene aerogel by the cysteamine vapor method. The graphene/cysteamine aerogel (GCA)-supported phase change material (PCM) composites can sustain their initial solid state during the phase transition process, even under external force. The pyro-electrode is composed of a conductor and pyro film that can produce electron movement via change of temperature difference between the two sides of the pyro-electrode. The indium tin oxide (ITO)-connected transparent pyro-electrode can transmit the sunlight to the PCM composite, and this pyroelectric generator provides the output electrical voltage and current upon the light-on/off process.

**Figure 2 materials-15-04541-f002:**
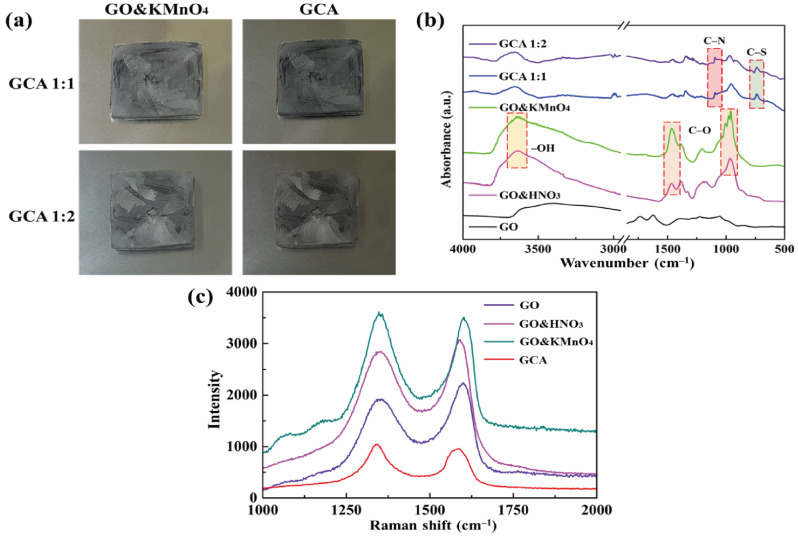
(**a**) Camera images of the oxidized graphene aerogel (GO&KMnO_4_) and graphene/cysteamine aerogel (GCA). (**b**) FTIR peaks of graphene oxide (GO), GO&HNO_3_, GO&KMnO_4_, GCA 1:1, and GCA 1:2. (**c**) Raman peaks of GO, GO&HNO_3_, GO&KMnO_4_, and GCA.

**Figure 3 materials-15-04541-f003:**
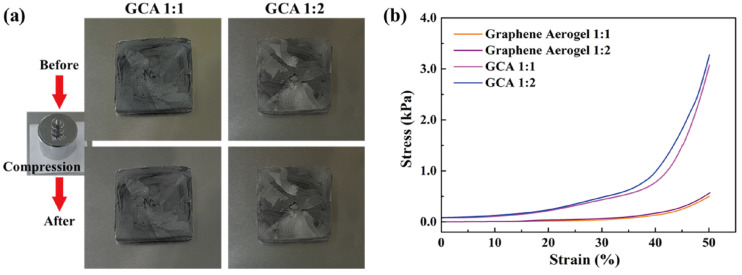
(**a**) Shape recovery test of GCA 1:1 and GCA 1:2. (**b**) Stress–strain tests of graphene aerogels and GCAs.

**Figure 4 materials-15-04541-f004:**
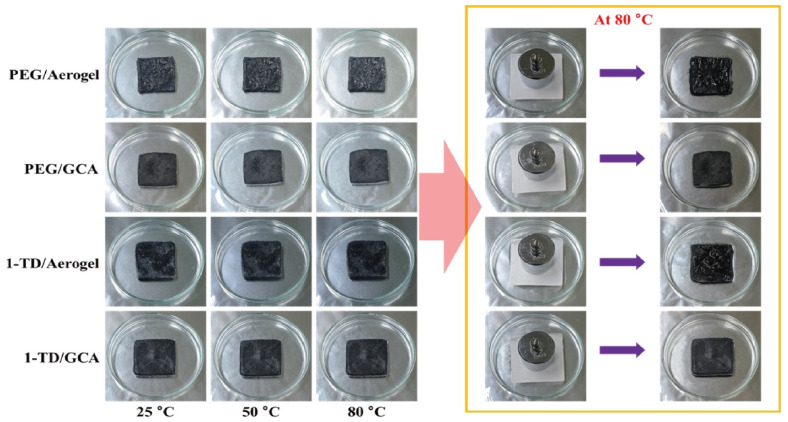
Form-stable test of the original aerogel and GCA-supported PCM composites.

**Figure 5 materials-15-04541-f005:**
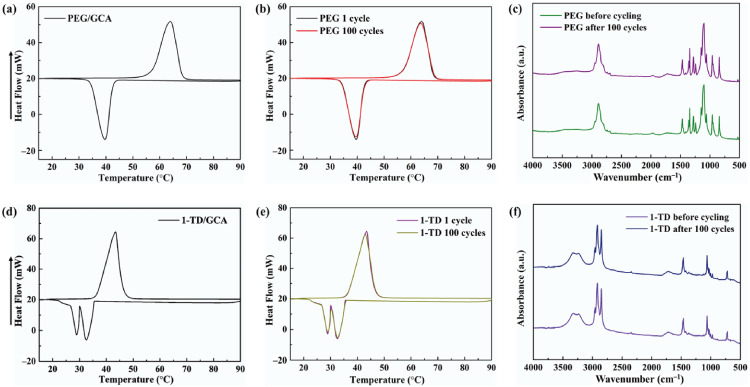
DSC curve of (**a**) the PEG/GCA composite and (**b**) DSC 100 cycling test. (**c**) FTIR peaks after 100 cycles. DSC curve of (**d**) 1-TD/GCA composite and (**e**) DSC 100 cycling test. (**f**) FTIR peaks after 100 cycles.

**Figure 6 materials-15-04541-f006:**
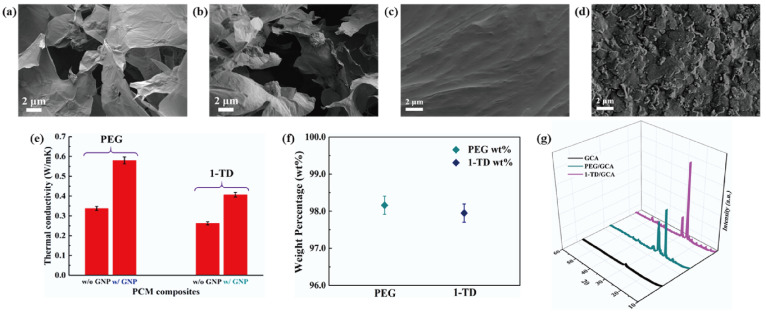
SEM images of (**a**) GCA 1:1, (**b**) GCA 1:2, (**c**) PEG/GCA composite, and (**d**) 1-TD/GCA composite. (**e**) Thermal conductivity of the PEG and 1-TD composites, and (**f**) the pure PCM weight fraction in the PCM composite. (**g**) XRD peaks of GCA, PEG/GCA, and 1-TD/GCA.

**Figure 7 materials-15-04541-f007:**
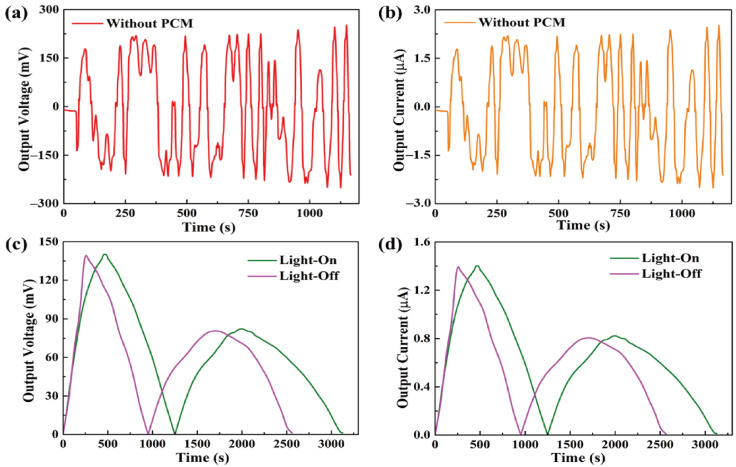
(**a**) Output voltage peak and (**b**) output current peak of the pyro-electrode without PCM. Solar light intensity of 15 mW/cm^2^, showing (**c**) the output voltage peaks during the light-on/off process and (**d**) the output current peaks during the light-on/off process.

**Figure 8 materials-15-04541-f008:**
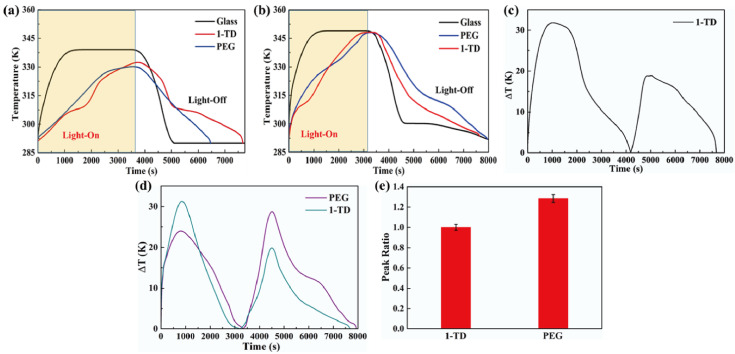
Temperature profiles of glass, 1-TD, and PEG under solar light (**a**) intensity of 10 mW/cm^2^ and (**b**) intensity of 15 mW/cm^2^. (**c**) Temperature difference of the 1-TD composite under 10 mW/cm^2^, (**d**) temperature difference of both the PEG and 1-TD composite under 15 mW/cm^2^, and (**e**) peak area ratios of the PCM composites under 15 mW/cm^2^.

**Figure 9 materials-15-04541-f009:**
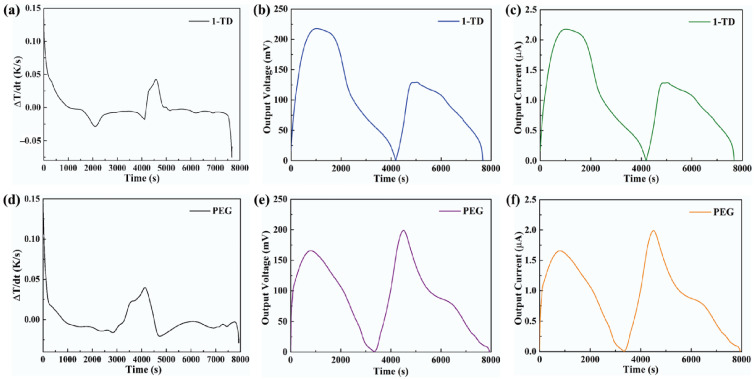
(**a**) Change of temperature difference by using the 1-TD composite under 10 mW/cm^2^, (**b**) output electrical voltage peak, and (**c**) output electrical current peak under 10 mW/cm^2^. Solar light intensity of 15 mW/cm^2^. (**d**) Change of temperature difference by using PEG composite, (**e**) output electrical voltage peak, and (**f**) output electrical current peak.

**Figure 10 materials-15-04541-f010:**
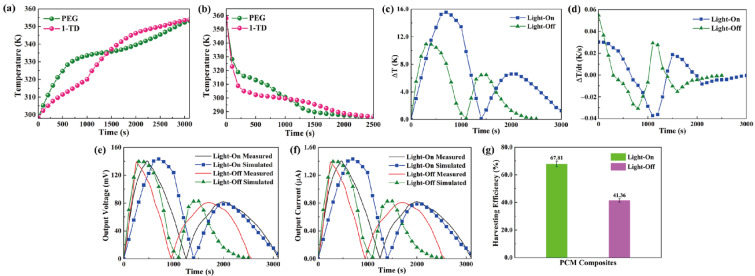
Multiple PCMs temperature profiles during the (**a**) light-on process and (**b**) light-off process. (**c**) Temperature difference upon the light-on/off process. (**d**) Change of temperature difference upon the light-on/off process. Experimental and simulation results of (**e**) the output voltage peaks upon the light-on/off process and (**f**) the output current peaks upon the light-on/off process. (**g**) Pyroelectric energy harvesting efficiencies during the light-on/off process.

**Figure 11 materials-15-04541-f011:**
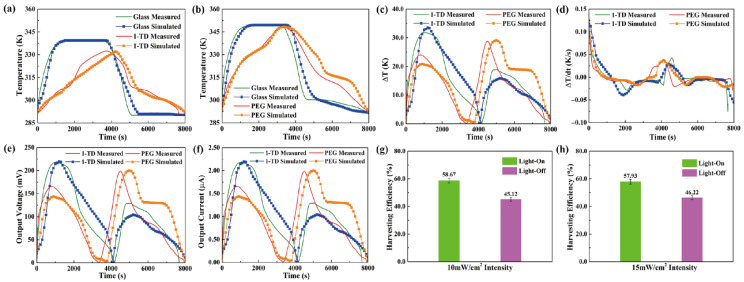
The experimental and simulation results of temperature peaks under (**a**) 10 mW/cm^2^ and (**b**) 15 mW/cm^2^ solar light intensity. (**c**) Temperature difference and (**d**) change of temperature difference in both the 1-TD and PEG composites under (10 and 15) mW/cm^2^ solar light intensity, respectively. (**e**) Output electrical voltage and (**f**) output electrical current in both the 1-TD and PEG composites under 10 and 15 mW/cm^2^ solar light intensity, respectively. Pyroelectric energy harvesting efficiency of (**g**) 10 mW/cm^2^ and (**h**) 15 mW/cm^2^ solar light intensity upon the light-on/off process.

**Table 1 materials-15-04541-t001:** Characteristics of the GCA recovery test.

Samples	Before Compression (cm)	After Compression (cm)	Recovery Time (s)	Recovery (%)
**GCA 1:1**	**0.5003**	**0.4996**	**0.52**	**99.86**
**GCA 1:2**	**0.5008**	**0.5000**	**0.53**	**99.84**

**Table 2 materials-15-04541-t002:** DSC results of the PEG/GCA and 1-TD/GCA composites.

Samples	T_m_ (°C)	ΔH_m_ (J/g)	T_c_ (°C)	ΔH_c_ (J/g)
**PEG/GCA**	**64.31**	**179.65**	**39.19**	**159.68**
**1-TD/GCA**	**42.76**	**214.01**	**30.42**	**211.94**
**PEG/GCA Cycling**	**64.11**	**179.26**	**39.03**	**159.41**
**1-TD/GCA Cycling**	**42.20**	**213.57**	**30.16**	**211.54**

## Data Availability

Data are made available upon reasonable request to the corresponding author.

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
