# Peer review of "Phase Change Material (PCM) Composite Supported by 3D Cross-Linked Porous Graphene Aerogel"

_materials, 2022, doi:10.3390/ma15134541_

Round 1

Reviewer 1 Report

In this paper, the authors reported that the modified three-dimensional porous graphene aerogel was used as a reliable support material to effectively reduce the volume shrinkage during the permeation process. This study briefly indicates a potential application of smart control and solar energy harvesting by using PCM composite. I believe that publication of the manuscript may be considered only after the following issues have been resolved.

1.      What are the advantages of this job? It is suggested that the author give a table to compare the advantages of this work in detail.

2.      In Figure 2a and Figure 3a, it is suggested that the author give a scale to mark the size of the sample.

3.      The text information in Figure 10 and Figure 11 is too small. The author is suggested to make adjustments.

4.      The introduction can be improved. The articles related to the some applications of phase change material should be added such as Phys. Chem. Chem. Phys., 2022, 24, 8846 – 8853. The articles related to the some applications of graphene should be added such as Electrochimica Acta 2020, 330, 135196; RSC Adv., 2022, 12(13), 7821-7829; ChemElectroChem, 6 (2019), pp. 5642-5650; Sens. Actuators B Chem. 2018, 260, 529–540.

5.      In Figure 2c, it is suggested that the author clearly mark the peak position.

Author Response

In this paper, the authors reported that the modified three-dimensional porous graphene aerogel was used as a reliable support material to effectively reduce the volume shrinkage during the permeation process. This study briefly indicates a potential application of smart control and solar energy harvesting by using PCM composite. I believe that publication of the manuscript may be considered only after the following issues have been resolved.

Response to Reviewer #1,

  1. What are the advantages of this job? It is suggested that the author give a table to compare the advantages of this work in detail.

We thank the referee for the positive review. In this work, we fabricated form-stable phase change material (PCM) composites for pyroelectric energy harvesting. The main advantage is the PCM composites constructed pyro-electrode can produce a stable and continuous electrical energy while many previous research referring to pyroelectric were not mentioned. From Figure 7, we can see without PCM connected pyro-electrode showed irregular output voltage and current but this work even using transparent pyro-electrode can generate usable electrical energy on light-on/-off process.

  1.  In Figure 2a and Figure 3a, it is suggested that the author give a scale to mark the size of the sample.

We thank the referee for the helpful suggestion. We fabricated 4cm×4cm size of graphene aerogel and added a brief description in this manuscript.

  1. The text information in Figure 10 and Figure 11 is too small. The author is suggested to make adjustments.

We thank the referee for the helpful comment. We added detail description in this manuscript. The Figure 10 and 11 are comparison between experimental and numerical results to demonstrate the possibility of pyro energy harvesting.

  1. The introduction can be improved. The articles related to the some applications of phase change material should be added such as Phys. Chem. Chem. Phys., 2022, 24, 8846 – 8853. The articles related to the some applications of graphene should be added such as Electrochimica Acta 2020, 330, 135196; RSC Adv., 2022, 12(13), 7821-7829; ChemElectroChem, 6 (2019), pp. 5642-5650; Sens. Actuators B Chem. 2018, 260, 529–540.

We thank the referee for the key point. We cited these references and modified detail in this section.

  1. In Figure 2c, it is suggested that the author clearly mark the peak position.

We thank the referee for the helpful suggestion. Some peaks were shifted slightly, and the increase of oxidation makes a little bit change these peak positions.

Reviewer 2 Report

In this paper a new platform of a pyro-electric power generator to produce solar-to-electrical energy harvesting was investigated.

The structure of the article fulfills the structure of a research article.

Five keywords (expressions) are included by the authors.

The Introduction section provides sufficient background information for readers in the immediate field to understand the problem that this study addresses.

In the Results and Discussion section, the authors present and interpret the results of the performed experiments.

The paper ends with the Conclusions part. In this section the authors mention the conclusions of their research study.

I suggest to Reconsider after Minor Revisions for the following reasons:

1.  The authors should highlight the novelty of the present work

2. Moderate English changes required

3. Use line and symbols in Figures because the graphs are difficult to be read on a black-white printer.

Author Response

In this paper a new platform of a pyro-electric power generator to produce solar-to-electrical energy harvesting was investigated. The structure of the article fulfills the structure of a research article.

Five keywords (expressions) are included by the authors.

The Introduction section provides sufficient background information for readers in the immediate field to understand the problem that this study addresses.

In the Results and Discussion section, the authors present and interpret the results of the performed experiments.

The paper ends with the Conclusions part. In this section the authors mention the conclusions of their research study.

I suggest to Reconsider after Minor Revisions for the following reasons:

Response to Reviewer #2,

  1. The authors should highlight the novelty of the present work

We thank the referee for the helpful comment. We modified the introduction with highlight the novelty in this manuscript.

  1. Moderate English changes required

We thank the referee for the helpful suggestion. We revised the English description in this manuscript.

  1. Use line and symbols in Figures because the graphs are difficult to be read on a black-white printer.

We thank the referee for the key point. We modified the figure effect in this manuscript and be read clearly by print. In addition, we can upload source files if were required.

Reviewer 3 Report

The paper presents the use of graphene aerogel supported by phase change material to control solar energy harvesting.

Researchers fabricated modified 3D structures of graphene aerogel and then performed material characterization and measurements of the output electrical energy during the light on/off process.
It should be emphasized that the authors received and tested the new platform of a pyroelectric power generator to fabricate solar-to-electrical energy harvesting.

Nevertheless, I have some comments and observations about the reviewed work.
1. There is no description of the method to obtain graphene aerogel.
2. How were the multiplate PCMs changed into the liquid phase?
3. How does the graphene cross-linking reaction occur in graphene aerogel (GCA) or are they physical, i.e. reversible?
4. Methodology for obtaining GCA1: 2 and GCA 1: 1 samples are not specified.
5. Lack of information in the methodology on how the mechanical properties tests were carried out, and what the sample for the mechanical properties tests looked like.
6. In my opinion, there is a lack of precise characteristics of the structure and dispersion of the nanofiller into the aerogel. The structure of the nanomaterial significantly affects its properties. Presented SEM images are insufficient to infer the structure.
7. Please verify the DSC plots, their graphical form is incorrect and the course suggests that PEG/GCA melting occurs at a temperature of approx. 40oC, while crystallization at approx. 65oC.

Author Response

Researchers fabricated modified 3D structures of graphene aerogel and then performed material characterization and measurements of the output electrical energy during the light on/off process.
It should be emphasized that the authors received and tested the new platform of a pyroelectric power generator to fabricate solar-to-electrical energy harvesting.

Nevertheless, I have some comments and observations about the reviewed work.

Response to Reviewer #3,

  1. There is no description of the method to obtain graphene aerogel.

We thank the referee for the helpful comment. We added the method of graphene aerogel in this manuscript. The KMnO4 oxidized graphene oxide (GO) with graphene nano-platelet (GNP) were dispersed in DI water and evaporated the solvent by freeze-drying method for over 48 h.

  1. How were the multiplate PCMs changed into the liquid phase?

We thank the referee for the key point. From the DSC result, we can see the melting point and that these phase change materials (PCMs) started to occur the solid-liquid phase transition by increasing of external temperature or absorbing the solar light.

  1. How does the graphene cross-linking reaction occur in graphene aerogel (GCA) or are they physical, i.e. reversible?

We thank the referee for the helpful suggestion. The cysteamine aqueous solution is placed at the vacuum chamber for 72 h at 150 °C. the cysteamine evaporation temperature is 135 °C that there were a lot of cysteamine gas moved into the oxidized graphene oxide (GO) aerogel skeletons. As a result, the cysteamine had a reaction with functional groups of GO aerogel to generate cross-linked chemical chains. This is a chemical reaction and not reversible.

  1. Methodology for obtaining GCA1: 2 and GCA 1: 1 samples are not specified.

We thank the referee for the helpful comment. We cited our previous research referring to methodology of two different GCAs as below:

Yu, C.; Kim, H.; Youn, J. R.; Song, Y. S., Enhancement of Structural Stability of Graphene Aerogel for Thermal Energy Harvesting. ACS Applied Energy Materials 2021

  1. Lack of information in the methodology on how the mechanical properties tests were carried out, and what the sample for the mechanical properties tests looked like.

We thank the referee for the helpful suggestion. The stress-strain curves were measured by using Universal Testing Machine (UTM) that compress the different graphene aerogels. To demonstrate the mechanical properties, the stress–strain curves were obtained to further demonstrate the increase of mechanical properties, as given in Fig. 3b. The samples were compressed at the ε = 50 % maximum strain, and showed that during the compression process, the initial graphene aerogels (GA) exhibited a weak mechanical property. The GCAs showed excellent mechanical properties with over 3.0 kPa stress, which indicated a significant increase in the cross-linked internal structures.

  1. In my opinion, there is a lack of precise characteristics of the structure and dispersion of the nanofiller into the aerogel. The structure of the nanomaterial significantly affects its properties.

Presented SEM images are insufficient to infer the structure.

We thank the referee for the key comment. The filler in this work was graphene nano-platelet (GNP) and this GNP exhibits high mechanical property that be dispersed with graphene oxide (GO) to prevent the crack of graphene aerogel. We cited our previous research about the structure of nanomaterial and SEM images as below:

Yu, C.; Youn, J. R.; Song, Y. S., Enhancement of Thermo-Electric Energy Conversion Using Graphene Nano-platelets Embedded Phase Change Material. Macromolecular Research 2021, 29, (8), 534-542

  1. Please verify the DSC plots, their graphical form is incorrect and the course suggests that PEG/GCA melting occurs at a temperature of approx. 40oC, while crystallization at approx. 65oC.

We thank the referee for the key suggestion. The figure 5a is correct. Based on the direction of heat flow, the upper peak was melting process. The PEG melting point is approximately 65 °C, and the cooling point is 40 °C. Because of high viscosity and high molecular weight, liquid PEG needs some part of stored energy to recover its initial solid state that crystallization process may not occur at 65 °C. Comparing with 1-TD, the PEG exhibits higher temperature difference between melting and cooling points.

Round 2

Reviewer 1 Report

Accept.

Reviewer 3 Report

I would like to thank the authors for clarifying my doubts related to the reviewed work.  The authors made changes to the work that significantly increase its scientific value of the work. In my opinion, the work in its current form is suitable for publication in the Materials journal.